

# Influence of purple non-sulfur bacterial augmentation on soil nutrient dynamics and rice (*Oryza sativa*) growth in acidic saline-stressed environments

Nguyen Quoc Khuong, Nguyen Minh Nhat, Le Thi My Thu and Le Vinh Thuc

College of Agriculture, Can Tho University, Ninh Kieu, Can Tho, Vietnam

## ABSTRACT

The aim of the current study was to assess the potency of the exopolymeric substances (EPS)-secreting purple non-sulfur bacteria (PNSB) on rice plants on acidic salt-affected soil under greenhouse conditions. A two-factor experiment was conducted following a completely randomized block design. The first factor was the salinity of the irrigation, and the other factor was the application of the EPS producing PNSB (*Luteovulum sphaeroides* EPS18, EPS37, and EPS54), with four replicates. The result illustrated that irrigation of salt water at 3–4‰ resulted in an increase in the $Na^+$ accumulation in soil, resulting in a lower rice grain yield by 12.9–22.2% in comparison with the 0‰ salinity case. Supplying the mixture of *L. sphaeroides* EPS18, EPS37, and EPS54 increased pH by 0.13, $NH_4^+$ by 2.30 mg $NH_4^+$ $kg^{-1}$, and available P by 8.80 mg P $kg^{-1}$, and decreased $Na^+$ by 0.348 meq $Na^+$ 100 $g^{-1}$, resulting in improvements in N, P, and K uptake and reductions in Na uptake, in comparison with the treatment without bacteria. Thus, the treatments supplied with the mixture of *L. sphaeroides* EPS18, EPS37, and EPS54 resulted in greater yield by 27.7% than the control treatment.

## INTRODUCTION

The Mekong Delta is the biggest rice-producing region in Vietnam, accounting for more than 50% of rice production and roughly 90% of exporting rice nationally. However, the salt water instrusion has been one of the main obstacles to rice cultivation in coastal areas, including Tra Vinh, Ben Tre, Soc Trang, Bac Lieu, Ca Mau, and Kien Giang provinces (*Hoan, Nguyen Khoi & Trung, 2019*; *Tan, Tran & Loc, 2020*; *Nam et al., 2022*; *Nhung et al., 2019*; *Poelma et al., 2021*; *Khuong et al., 2021*). Bac Lieu is one of the three provinces that are affected by salt water instruction the most, leading to 5,400 ha of soil that lacks fresh water, yield reduction, and increases in cost to prevent the damage (*Ministry of Agriculture and Rural Development, 2020*). Consequently, many farmers alienate a cultivation system of integrated rice and shrimp; shrimp is raised in the dry season and rice in the wet season (*Kruse et al., 2020*). This cultivation system is considered to be sustainable in the aspects of

Corresponding author
Le Vinh Thuc, lvthuc@ctu.edu.vn

economy, environment, and adaptability to climate changes (*Braun et al., 2019*; *Dang, 2020*).

In rice cultivation, in order to reduce damages caused by salt effects, approaches, including applying lime, sludge from ditches, organic amendments, biochar, and salt-tolerant rice cultivars, are considered to be popular and efficient (*Minh et al., 2019*; *Kabir et al., 2020*; *Litardo et al., 2022*; *Haque et al., 2021*; *Phuong et al., 2020*). However, in order to achieve the sustainable agriculture, chemicals need to be cut off and altered by biological approaches (*Shultana et al., 2020a*). The purple non-sulfur bacteria (PNSB) are highly potent due to their ability to live in different conditions and to tolerate adverse environments, such as high salinity (*Khuong et al., 2021*; *Nunkaew et al., 2015a*; *Panwichian et al., 2010*). Therefore, they can live as a phototroph when being submerged on paddy fields, which makes them a potent candidate for ameliration of rice cultivation. Moreover, they also contribute to greater cultivating effectiveness on rice planted on salt-affected soil (*Nunkaew et al., 2014*; *Kantachote et al., 2016*; *Kantha, Kantachote & Klongdee, 2015*). This can be interpreted that the PNSB are capable of fixing nitrogen (N), and solubilizing insoluble phosphate (P) and potassium (K) for better nutrient uptake of the plants (*Khuong et al., 2018*, *2020*, *2021*; *Sakpirom et al., 2017*, *2021*). Besides, PNSB are capable of producing the phytohormones indole-3-acetic acid (IAA), siderophores, and 5-aminolevulinic acid (ALA) that can promote plant growth (*Khuong et al., 2020*, *2021*; *Sakpirom et al., 2017*). In particular, PNSB secrete exopolymeric substances (EPS) that are able to bind $Na^+$ ion in order to reduce saline level and enhance plant growth, which minimizes the adverse effects of salinity (*Khuong et al., 2021*; *Nunkaew et al., 2015b*). This is because main components of EPS are polysaccharides, containing functional groups, which can immobilize $Na^+$ (*Isfahani et al., 2018*; *Shultana et al., 2020b*; *Nunkaew et al., 2015a*). Moreover, EPS can have various functions, such as adhension, cell aggregation, cohension, electron acceptor and donor, energy source, barrier, sorption, water retention, *etc* (*Siddharth et al., 2021*). Nowadays, although the PNSB *L. sphaeroides* species, which can provide EPS, have been isolated from saline soil of a cultivation system of integrated rice and shrimp, the efficiency of these strains has not been evaluated on rice cultivation yet. Moreover, using EPS producing bacteria to lessen the damage caused by salinity and to improve the growth and yield of rice plants is one of the trending orientation in sustainable agriculture. Therefore, the study was aimed to identify the potency of the strains of EPS-producing *L. sphaeroides* on improving the properties of acidic salt-affected soil and reducing the damage in rice yield on that soil. In another word, the study was working on: How would the EPS-producing bacteria affect the salinity in the experiment soil? Would the EPS-producing bacteria be efficient in promoting rice growth and yield, and in reducing saline stress on rice plants that were being damaged by the salinity from the soil and the water?

## MATERIALS AND METHODS

The experiment was conducted on acidic salt-affected soil at the surface level (0–30 cm) in Hong Dan district, Bac Lieu province (Fig. S1). The EPS-producing PNSB isolated from

soil of a cultivation system of integrated rice and shrimp were used, and they were the best in producing EPS. They were isolated from saline soil system with their identification of *Luteovulum sphaeroides* EPS18, EPS37, and EPS54, with their accession numbers as OR044031, OR044032 and OR044033, respectively. These bacteria were chosen in our preliminary data for being isolated from a saline ecosystem and being able to produce EPS, which is known as a mechanism to reduce saline stress on rice as mentioned in the introduction. The fertilizers in the current study consisted of Urea (46% N), super phosphate (16% $P_2O_5$) and potassium (60% $K_2O$).

## Experimental design

The two-factor experiment followed a completely randomized block design with 20 treatments and four replicates, each of which was a pot of soil. In detail, factor (A) was the salinity of the water (0‰, 2‰, 3‰, and 4‰) and factor (B) was the application of the EPS-producing PNSB (negative control; supplied with the single strain of EPS18, EPS37, or EPS54; and supplied with the mixture of the three strains EPS18, EPS37, and EPS54) (Fig. S2). The experiment was conducted at the greenhouse (10.029783 N, 105.767414 E) in the Research and Experiment Agriculture Station, College of Agriculture, Can Tho University from June 2022 to November 2022, which was the fall-winter rice cropping season, one of the two major rice cropping seasons in Vietnam. In the greenhouse, the light and dark hours per day were 11 and 13, respectively; the temperature was 36 °C and humidity was 60%.

## Soil preparation

The soil from the cultivation system of integrated rice and shrimp was collected, dried naturally, smashed, removed from plant residues. Eight kg of dry soil was put into a pot whose size at large bottom diameter × small bottom diameter × height was 23 × 17 × 18 cm (80 pots in total), respectively, and added with 2 L of water into each pot to make the soil saturated in waterlogged rice cultivation. The properties of experimental soil was characterized in Table 1.

## Seed treatment and sowing

The rice seeds were sterilized with ethanol and sodium hypochlorite (1%) and then rinsed with deionized water to ensure the seeds were fully sterilized. Next, the seeds were incubated to germinate for 24 h in the dark. Then, 500 rice seeds were divided into five equal portions into five beakers (100 seeds in each beaker). Four beakers contained 10 mL of bacterial suspension ($1 \times 10^9$ CFU $mL^{-1}$) of EPS18, EPS37, and EPS54 and their mixture for 1 h before sowing, and 1 beaker contained distilled water as the control treatment. Each pot was sowed with six seeds per pot, possessing a density of $1.2 \times 10^4$ CFU $g^{-1}$ dry soil weight (DSW) to suit the amount of 8 kg of soil for soil not to be the restricitng parameter. Each treatment was performed with four replicates. Each treatment contained 24 seeds to meet the requirement for pot experiment as the guidance of the International Rice Research Institute (*International Rice Research Institute, 2014*).

Table 1 The properties of the soil at the beginning of the crop at the depth of 0–20 cm in Hong Dan-Bac Lieu.

| Properties | Unit | Value |
|---|---|---|
| $pH_{H2O}$ | - | $3.14 \pm 0.11$ |
| $pH_{KCl}$ | - | $2.70 \pm 0.08$ |
| EC | mS cm$^{-1}$ | $5.08 \pm 0.14$ |
| $N_{total}$ | %N | $0.82 \pm 0.03$ |
| $NH_4^+$ | mg kg$^{-1}$ | $117.2 \pm 5.62$ |
| $P_{total}$ | %$P_2O_5$ | $0.022 \pm 0.001$ |
| $P_{available}$ | mg kg$^{-1}$ | $10.2 \pm 0.54$ |
| Al-P | mg kg$^{-1}$ | $16.9 \pm 1.22$ |
| Fe-P | mg kg$^{-1}$ | $54.7 \pm 3.16$ |
| Ca-P | mg kg$^{-1}$ | $21.3 \pm 1.22$ |
| CEC | meq 100 g$^{-1}$ | $6.89 \pm 0.44$ |
| $Na^+$ | meq 100 g$^{-1}$ | $1.74 \pm 0.09$ |
| $K^+$ | meq 100 g$^{-1}$ | $0.50 \pm 0.05$ |
| $Ca^{2+}$ | meq 100 g$^{-1}$ | $2.44 \pm 0.13$ |
| $Mg^{2+}$ | meq 100 g$^{-1}$ | $9.07 \pm 1.82$ |

Note:
Values are mean ± standard deviation of four replications. EC, electrical conductivity; CEC, cation exchange capacity.

## Fertilizers

The fertilization following the recommendation included 100 N–60 $P_2O_5$–30 $K_2O$ (kg ha$^{-1}$) (Tan & Hach, 2013). The P fertilizer was applied 100% three days before sowing, and the N fertilizer was utilized at the rates of 30%, 30%, and 40% at 10, 20, and 45 days after sowing (DAS). The K fertilizer was applied in halves at 10 and 45 DAS.

## Watering

Tap water was kept from 3 to 5 cm height from the soil during the experiment, because the paddy rice cultivars require a submerging condition. Moreover, in the saline water irrigated treatments, 5 mL of saline water was poured into each pot at 20, 27, 34, 41, 48, and 55 DAS to maintain a sustainable saline stress on rice and bacteria there, while control was the tap water.

## Bacterial inoculation

A total of 3 mL of $1 \times 10^9$ CFU mL$^{-1}$ bacteria (for the mixed PNSB, each strain accounted for 1 mL) was applied three times at 13, 18, and 30 DAS, to maintain an acceptable density of the bacteria before applying saline water. However, further number of PNSB applications was not suitable for farmers to conduct. Thus, the supplied amount of bacteria was $1.8 \times 10^6$ CFU g$^{-1}$ DSW, and the total amount from both seeds and liquid biofertilizers was $1.812 \times 10^5$ CFU g$^{-1}$ dry soil. The functions of PNSB were to provide EPS (1.82–1.97 and 1.14–1.59 mg L$^{-1}$), to fix N (4.16–4.69 and 7.36–9.65 mg L$^{-1}$), and to solubilize P from Ca-P (11.8–13.7 and 15.6–18.9 mg L$^{-1}$), to dissolve K (1.56–2.16 and 1.44–1.73 mg L$^{-1}$),

and to produce ALA (2.14–2.56 and 2.96–3.16 mg $L^{-1}$) under microaerobic light and aerobic dark conditions, respectively.

## Soil analysis

At the beginning and at harvest, the soil was drilled, collected, dried up, then removed from plant residues, smashed, and filtered *via* a sieve of 0.5 and 2.0 mm. All the soil analytic techniques were composed by *Sparks et al. (1996)* and briefly summarized as follows: $pH_{H2O}$ or $pH_{KCl}$ was extracted with a ratio of soil:water (1:2.5) or soil:1.0 M KCl (1:2.5) and measured by a pH meter. The solution derived from $pH_{H2O}$ measurement was utilized for determining the EC value by an EC meter. The total N was digested by a mixture of saturated $H_2SO_4$-$CuSO_4$-Se, with a ratio of 100-10-1, determined by the Kjeldahl method, and titrated by 0.01 N $H_2SO_4$. The available N was measured by the blue phenol method at the 640 nm wavelength. The total P was converted into inorganic forms by a mixture of saturated $H_2SO_4$-$HClO_4$ and revealed in color by ascorbic acid at the 880 nm wavelength. The available P was determined by soil extraction with 0.1 N $HCl+0.03$ N $NH_4F$, where the soil : water ratio was 1:7. Then, ascorbic acid was utilized for exposing colors at 880 nm wavelength. The insoluble forms of P comprising of ferrous phosphate (Fe-P), aluminum phosphate (Al-P) and calcium phosphate (Ca-P), were extracted by corresponding solutions, including 0.10 M NaOH, 0.50 M $NH_4F$, and 0.25 M $H_2SO_4$, determined by ascorbic acid. Their colors were compared using a spectrophotometer at the 880 nm wavelength. The cation exchange capacity (CEC) was measured by extracting soil with 0.1 M $BaCl_2$ and titrated by 0.01 M EDTA. The concentrations of $K^+$, $Na^+$, $Ca^{2+}$, and $Mg^{2+}$ from the CEC extracted solution were measured using a spectrophotometer at wavelengths of 766.5, 589, 422.7, and 285.2 nm, respectively.

## Bacterial densities

The parameter in the soil was determined according to the Most Probable Number (MPN) method of *Naoki, Nishiyama & Matsumoto (2001)*. In brief, soil after diluted at $10^{-6}$ was spread on a dish, which was incubated in an anaerobic jar with gas-pak in 7 days. After incubation, the colonies appearing on the dish were counted.

## Biochemical properties of the rice

The leaves and stem (straw) of rice were collected at 45 DAS, which is the stage that rice plants fully develop their leaves, in order to analyze proline by the ninhydrin method of *Bates, Waldren & Teare (1973)*, which is briefly summarized as follows: the humidity of the samples was measured by drying up the samples of straw of rice seedling at 70 °C for 72 h. Then, 0.5 g of the rice samples was transferred to a tube whose size was $13 \times 100$ mm. A total of 10 mL of sulfosalicylic acid 3% was added to completely mill the samples, which were then shaken for 30 min and centrifuged at 3,000 rpm for 15 min. A clear solution was finally collected, and the debris was removed. The colorimetric analysis was conducted by reacting with 2.0 mL of the sample solution with 2.0 mL of ninhydrin and 2 mL of glacial acetic acid. The complex was well mixed and capped. The incubation was applied for 1 h at

100 °C. After that, the ice was used to cool it down, and 4 mL of toluene was added to the extracted solution, which was then shaken for 15–20 s and measured using a spectrophotometer at the 520 nm wavelength.

## The concentrations of N, P, K, and Na in plants

The analysis of the concentrations of N, P, K, and Na in rice at harvesting (90 DAS) was conducted according to the method of *Walinga et al. (1989)*. Rice straw and seeds of a pot at harvest were weighed, dried at 70 °C for 72 h, and measured for their dry weight. Then, they were then carefully milled and utilized for analyzing the concentrations of N, P, K, and Na in straw, and seeds. Samples were digested by a mixture of saturated $H_2SO_4$ and salicylic acid. The concentration of N in the sample was measured by the Kjeldahl method (Velp UDK129). The P was determined at 880 nm wavelength by a spectrophotometer (UV-VIS 1800; Shimadzu, Kyoto, Japan). Finally, K and Na were measured using an atomic spectrophotometer ICP–OES (Icap 6300 Duo; Thermo Fisher Scientific, Waltham, MA, USA).

## Evaluating the growth, yield components, and grain yield

All the growth and agronomic parameters were measured following the description of *International Rice Research Institute (2014)*.

The growth was measured by plant height and panicle length. The plant height of rice and their panicle length were checked at 90 DAS (harvesting). The plant height was determined from the ground to the peak leaf or the peak panicle, and four plants of a pot were put into measurement. The panicle length was determined from the neck of a panicle to its end, and eight panicles of a pot were checked.

### The yield components

The number of panicles per pot: the number of panicles of a pot was counted. The number of seeds per panicle: the number of seeds of eight panicles per pot was counted. The percentage of filled seeds was the total number of filled seeds per the number of seeds × 100 (%). A total of 1,000-seed weight: 1,000 filled seeds in each treatment were weighed. The actual yield: the weight and humidity of seeds at harvest were measured in a pot and converted into that at 14% humidity.

## Statistical analysis

The data was processed and analyzed with two-way ANOVA by SPSS software, version 13.0. The Duncan *post-hoc* test was applied to compared the differences between means of treatments at 5% significance.

## RESULTS

### The influences of PNSB and the saline irrigation on soil fertility

The pH extracted by water values in the treatments irrigated with saline water at 2‰, 3‰, and 4‰ concentration differed at 5% significance ($p = 0.019$), in comparison with that in the treatment without saline irrigation, with 3.77 compared to 3.95. On the other hand, the $pH_{H2O}$ in the treatments supplied with a single strain of EPS18 or EPS37 or the PNSB

Table 2 The influence of the water salinity and the supplementation of exopolymeric subtances providing purple non-sulfur bacteria on the salt-contaminated Hong Dan-Bac Lieu soil properties in greenhouse condition.

| Factors | | $pH_{H2O}$ | $pH_{KCl}$ | EC (mS cm$^{-1}$) | CEC (meq 100 g$^{-1}$) | Na$^+$ | K$^+$ | Mg$^{2+}$ | Ca$^{2+}$ |
|---|---|---|---|---|---|---|---|---|---|
| | | – | – | | | | | | |
| Water salinity (A) (‰) | 0 | 3.77 ± 0.06[b] | 3.27 ± 0.04 | 0.75 ± 0.08[b] | 14.2 ± 1.97 | 0.835 ± 0.064[b] | 0.154 ± 0.029 | 3.114 ± 0.61 | 0.244 ± 0.03[b] |
| | 2 | 3.87 ± 0.15[ab] | 3.29 ± 0.10 | 0.76 ± 0.12[b] | 14.3 ± 2.01 | 0.881 ± 0.065[b] | 0.150 ± 0.016 | 3.007 ± 0.62 | 0.237 ± 0.06[b] |
| | 3 | 3.93 ± 0.12[a] | 3.32 ± 0.06 | 0.82 ± 0.15[b] | 14.3 ± 1.65 | 0.968 ± 0.101[a] | 0.160 ± 0.018 | 3.215 ± 0.35 | 0.277 ± 0.04[a] |
| | 4 | 3.90 ± 0.07[a] | 3.32 ± 0.07 | 0.93 ± 0.09[a] | 14.9 ± 1.76 | 0.956 ± 0.070[a] | 0.143 ± 0.024 | 3.009 ± 0.28 | 0.266 ± 0.03[ab] |
| PNSB (B) (1.812 × 10$^5$ CFU g$^{-1}$ dry soil) | NAPNSB | 3.75 ± 0.06[b] | 3.33 ± 0.06 | 0.96 ± 0.12[a] | 14.9 ± 1.49 | 1.144 ± 0.059[a] | 0.132 ± 0.026[c] | 3.066 ± 0.53 | 0.273 ± 0.04 |
| | EPS18 | 3.92 ± 0.09[a] | 3.30 ± 0.11 | 0.82 ± 0.12[b] | 14.5 ± 1.89 | 0.837 ± 0.071[ab] | 0.156 ± 0.024[ab] | 3.085 ± 0.41 | 0.251 ± 0.04 |
| | EPS37 | 3.95 ± 0.21[a] | 3.30 ± 0.06 | 0.82 ± 0.09[b] | 14.7 ± 1.93 | 0.875 ± 0.084[b] | 0.145 ± 0.017[bc] | 3.288 ± 0.66 | 0.263 ± 0.06 |
| | EPS54 | 3.83 ± 0.09[ab] | 3.27 ± 0.06 | 0.80 ± 0.16[b] | 13.3 ± 1.63 | 0.896 ± 0.100[b] | 0.158 ± 0.020[ab] | 2.911 ± 0.27 | 0.249 ± 0.03 |
| | MTPNSB | 3.88 ± 0.06[a] | 3.28 ± 0.05 | 0.68 ± 0.08[c] | 14.5 ± 2.31 | 0.796 ± 0.060[c] | 0.167 ± 0.023[a] | 3.081 ± 0.45 | 0.244 ± 0.04 |
| F (A) | | * | ns | * | * | * | ns | ns | * |
| F (B) | | * | ns | * | * | * | * | ns | ns |
| F (A × B) | | * | * | * | * | * | ns | ns | ns |
| CV (%) | | 3.88 | 2.46 | 16.1 | 16.1 | 9.16 | 16.0 | 19.4 | 18.5 |

| Factors | | Total N (%) | $NH_4^+$ (mg kg$^{-1}$) | Total P (%) | Available P (mg kg$^{-1}$) | Al-P | Fe-P | Ca-P | Log PNSB (MPN g$^{-1}$ DSW) |
|---|---|---|---|---|---|---|---|---|---|
| Water salinity (A) (‰) | 0 | 0.145 ± 0.02 | 17.2 ± 2.19 | 0.049 ± 0.004 | 42.4 ± 3.0 | 62.5 ± 4.0 | 192.6 ± 15.1 | 56.2 ± 5.3 | 5.19 ± 0.13[a] |
| | 2 | 0.158 ± 0.01 | 18.0 ± 1.77 | 0.051 ± 0.005 | 42.8 ± 2.9 | 64.2 ± 6.2 | 197.5 ± 15.4 | 57.0 ± 3.3 | 5.02 ± 0.12[b] |
| | 3 | 0.159 ± 0.02 | 17.8 ± 0.93 | 0.053 ± 0.005 | 42.1 ± 4.1 | 65.3 ± 2.5 | 191.1 ± 16.9 | 57.7 ± 3.7 | 4.91 ± 0.14[c] |
| | 4 | 0.148 ± 0.02 | 17.8 ± 1.26 | 0.052 ± 0.003 | 43.3 ± 5.1 | 61.6 ± 3.9 | 194.6 ± 17.2 | 57.2 ± 6.7 | 4.73 ± 0.07[d] |
| PNSB (B) (1.812 × 10$^5$ CFU g$^{-1}$ dry soil) | NAPNSB | 0.148 ± 0.01 | 15.6 ± 1.04[b] | 0.054 ± 0.005 | 35.6 ± 3.4[d] | 67.5 ± 2.7[a] | 200.7 ± 15.4[a] | 69.4 ± 4.7[a] | 2.11 ± 0.14[d] |
| | EPS18 | 0.155 ± 0.02 | 18.5 ± 1.74[a] | 0.049 ± 0.006 | 39.8 ± 4.6[c] | 62.2 ± 2.3[b] | 195.6 ± 17.2[a] | 54.3 ± 1.8[b] | 5.57 ± 0.12[b] |
| | EPS37 | 0.159 ± 0.02 | 17.7 ± 2.22[a] | 0.051 ± 0.004 | 51.1 ± 3.5[a] | 61.1 ± 2.5[b] | 197.9 ± 17.9[a] | 55.0 ± 5.9[b] | 5.55 ± 0.08[bc] |
| | EPS54 | 0.147 ± 0.03 | 18.6 ± 1.74[a] | 0.051 ± 0.003 | 42.3 ± 1.8[bc] | 66.8 ± 9.5[a] | 198.1 ± 19.4[a] | 57.5 ± 5.5[b] | 5.47 ± 0.15[c] |
| | MTPNSB | 0.151 ± 0.02 | 17.9 ± 0.95[a] | 0.051 ± 0.003 | 44.4 ± 5.5[b] | 59.4 ± 3.8[b] | 177.4 ± 10.9[b] | 48.9 ± 5.8[c] | 6.13 ± 0.07[a] |
| F (A) | | ns | ns | ns | ns | ns | ns | ns | * |
| F (B) | | ns | * | ns | * | * | * | * | * |
| F (A × B) | | * | ns | ns | ns | * | * | * | ns |
| CV (%) | | 15.4 | 10.2 | 10.6 | 12.2 | 9.81 | 9.24 | 9.54 | 2.45 |

Note:
In the same column, numbers followed by different letters were different from each other at 5% significance (*), ns, not significant; NAPNSB, no application of purple non-sulfur bacteria; EPS18, application of EPS18 strain; EPS37, application of EPS37 strain; EPS54, application of EPS54 strain; MTPNSB, application of the mixture of strains EPS18, EPS37 and EPS54. EC, electrical conductivity; CEC, cation exchange capacity.

mixture was 3.92, 3.95, and 3.88, respectively, while that in the treatment without bacteria was only 3.75. In the meantime, the $pH_{KCl}$ did not change significantly under the influence of the saline water and the bacteria addition (Table 2). However, there were significant interactions between the two factors influencing on both $pH_{KCl}$ and $pH_{H2O}$ (Table S1). In detail, the $pH_{H2O}$ increased in the treatments supplied with the single EPS37 strain at 2–3‰ salinity and with the single EPS54 strain at 3‰ salinity, in comparison to other salinity levels. For the $pH_{KCl}$, an identical trend as the $pH_{H2O}$ was observed and recorded in Table S2.

The treatments irrigated with saline water had EC values that were significantly different from that in the treatment without saline water irrigation at 5% significance ($p = 0.0014$); that is, the values of 0.75, 0.76, 0.82, and 0.93 mS cm$^{-1}$ correspond to the salinity of 0‰, 2‰, 3‰, and 4‰. In addition, in the treatments supplied with the bacteria, the EC changed significantly at 5% ($p = 0.000064$) from that in the treatment without bacteria. To be more specific, the treatments supplied with a single strain of either EPS18, EPS37, or EPS54 or their mixture reduced the EC value, in comparison with that in the treatment without bacteria supplied, with 0.68–0.82 mS cm$^{-1}$ compared to 0.96 mS cm2$^{-1}$. Among those treatments, the treatment supplied with the mixture of EPS18, EPS37, and EPS54 reduced the EC the most (Table 2).

The total N and P contents did not differ significantly among the levels of saline water and the supplementation of the PNSB. However, for the available N content, although the salinity of the water resulted in equivalent outcomes of the available N concentration (17.7 mg NH$_4^+$ kg$^{-1}$), in the treatments supplied with the PNSB individually or in mixture, the available N content was roughly 17.9–18.6 mg NH$_4^+$ kg$^{-1}$, which was greater than that in the treatment without bacteria (15.6 mg NH$_4^+$ kg$^{-1}$) (Table 2). For the available P content, the treatments supplied with a single strain of EPS18, EPS37, and EPS54 or their mixture had the available P content of 39.8–51.1 mg P kg$^{-1}$, remarkably greater than the available P content in the treatment without bacteria (35.6 mg P kg$^{-1}$). The concentrations of Al-P, Fe-P, and Ca-P were not statistically different at saline levels of the water. However, in the treatment supplied with the PNSB mixture, the concentrations of Al-P, Fe-P, and Ca-P were 59.4, 177.4, and 48.9 mg P kg$^{-1}$, respectively, and were lower at than those in the treatment without bacteria ($p = 0.0044$, $p = 0.0519$, $p = 0.00003$). Furthermore, although the treatments supplied with an individual strain of EPS18 and EPS37 reduced the Al-P content, supplying a single strain of EPS18, EPS37, and EPS54 did not contribute to the reduction of Fe-P content, with 195.6–198.1 compared to 200.7 mg P kg$^{-1}$ in the treatment without bacteria. Meanwhile, applying an individual strain of them reduced the content of Ca-P, with 54.3–57.5 compared to 69.4 mg P kg$^{-1}$ (Table 2).

CEC and the concentration of Mg$^{2+}$ were not statistically different among the levels of saline water and the supplementation of the PNSB. Increasing the salinity of the water resulted in a greater concentration of Na$^+$ accumulated in soil, with salt concentration of 0‰, 2‰, 3‰, and 4‰ corresponding to 0.835 ~ 0.811 > 0.968 ~ 0.956 meq Na$^+$ 100 g$^{-1}$. However, the treatments supplied with a single strain of EPS37 and EPS54 reduced the content of Na$^+$, compared with the control treatment. Noticeably, the treatment supplied with the mixture of EPS18, EPS37, and EPS54 had the lowest amount of Na$^+$ (0.796 meq Na$^+$ 100 g$^{-1}$). Moreover, in this parameter, the two factors significantly interacted. To be more specific, the highest Na$^+$ contents were in the treatments without bacteria at 2–4‰ salinity (1.150–1.250 meq Na$^+$ 100 g$^{-1}$), while the lowest one was found in the treatment with only the bacterial mixture (0.643 meq Na$^+$ 100 g$^{-1}$). Moreover, at the same salinity, even at 0‰ salinity, treatments with bacteria all resulted in lower concentration of Na$^+$ in soil, except for the treatment with the single EPS18 strain at 0‰ salinity. For the content of K$^+$, the difference was insignificant ($p = 0.169$) among saline levels of the water, approximately 0.143–0.160 meq K$^+$ 100 g$^{-1}$. However, adding a single strain of EPS18 and

EPS54 or the PNSB mixture led to a rise in the $K^+$ content by 0.024–0.026 meq $K^+$ 100 $g^{-1}$ compared with that in the treatment without bacteria. For the amount of $Ca^{2+}$, in the treatments applied with 3–4‰ saline water, the concentration of $Ca^{2+}$ was greater than that in the treatment with no saline water. However, the $Ca^{2+}$ content did not change significantly, and was 0.256 meq $Ca^{2+}$ 100 $g^{-1}$ on average (Table 2).

The increasing levels of salinity of the water resulted in a reduction in the density of PNSB, with 5.19 > 5.02 > 4.91 > 4.73 MPN $g^{-1}$ DSW at salinity of 0‰, 2‰, 3‰, and 4‰, respectively. Nevertheless, the supplementation of the bacteria increased the bacterial density. In detail, the bacterial density peaked in the treatment supplied with the mixture of the three PNSB strains (6.13 MPN $g^{-1}$ DSW), and the treatment without bacteria had the lowest bacteria density (2.11 MPN $g^{-1}$ DSW) (Table 2).

## The influences of the PNSB and the saline irrigation on the proline production in rice

The concentration of proline increased along with the salinity of the water from 0‰, 2‰, 3‰ to 4‰, with a value of 3.64 < 3.96 < 4.87 < 5.28 µmol $g^{-1}$ DW. Furthermore, in the treatments supplied with PNSB, the proline content was lower than that in the treatment without bacteria. To be more specific, the accumulation of proline in the treatments supplied with the PNSB individually or in mixture was 4.02–4.32 µmol $g^{-1}$ DW, while that in the treatment without bacteria was 5.60 µmol $^{-1}$ DW (Table 3).

## The influences of the PNSB and the saline irrigation on the biomass, the N, P, K, and Na concentration, and uptake of rice

### Dry biomass

Elevating the concentration of salt in the water from 2‰, 3‰ to 4‰ resulted in a reduction in dry biomass from 27.0 > 25.0 > 23.3 > 22.2 g $pot^{-1}$ in straw and 17.2 > 15.5 > 14.4 > 13.6 g $pot^{-1}$ in seeds. However, supplying the PNSB individually or in mixture had the dry biomass in straw, and seeds greater than those in the treatments without bacteria, except for the case of supplying the strain of EPS18 for straw dry biomass (Table 3). Moreover, significant interactions were found between the PNSB supplementation and the saline irrigation. In particular, the treatment with only the bacterial mixture and one without bacteria at 4‰ salinity resulted in the highest and the lowest dry biomass in straw, respectively. At 0‰ salinity, the treatments with PNSB resulted in the greater dry biomass in straw, except for supplying the single EPS37 strain, compared with the control treatment. At 2–3‰ salinity, only the mixture of the PNSB statistically improved the dry biomass in straw. In the meantime, in seeds, at every salinity, treatments with the PNSB all resulted in greater dry biomass than those of the treatments without bacteria.

### N, P, K, and Na concentration

#### N content

According to Table 3, when irrigating the 2‰ saline water, the N concentration in straw was 1.220%, equivalent to that in the treatments without applied saline water (1.168%). However, at 3‰ salinity, the N content in straw dropped to 1.081%. Meanwhile, the
**Table 3 The influence of the water salinity and the supplementation of exopolymeric subtances providing purple non-sulfur bacteria on the proline content, the dry biomass, the concentrations of N, P, K and Na in stem, leaves and in seeds of rice cultivated on the salt-contaminated Hong Dan-Bac Lieu soil in greenhouse conditions.**

| Factors | | Proline ($\mu$mol g$^{-1}$) | Dry biomass (g po$^{-1}$) | |
| --- | --- | --- | --- | --- |
| | | | Stem, leaves | Seeds |
| The water salinity (A) (‰) | 0 | $3.64 \pm 0.29^{d}$ | $27.0 \pm 1.14^{a}$ | $17.2 \pm 0.72^{a}$ |
| | 2 | $3.96 \pm 0.31^{c}$ | $25.0 \pm 1.40^{b}$ | $15.5 \pm 0.43^{b}$ |
| | 3 | $4.87 \pm 0.45^{b}$ | $23.3 \pm 1.71^{c}$ | $14.4 \pm 0.86^{c}$ |
| | 4 | $5.28 \pm 0.33^{a}$ | $22.2 \pm 1.50^{d}$ | $13.6 \pm 0.70^{d}$ |
| PNSB (B) (1.812 × 10$^{5}$ CFU g$^{-1}$ dry soil) | NAPNSB | $5.60 \pm 0.24^{a}$ | $22.5 \pm 1.64^{c}$ | $12.0 \pm 0.45^{c}$ |
| | EPS18 | $4.32 \pm 0.37^{b}$ | $22.8 \pm 0.88^{c}$ | $15.3 \pm 0.74^{b}$ |
| | EPS37 | $4.18 \pm 0.27^{b}$ | $24.0 \pm 1.12^{b}$ | $15.7 \pm 0.54^{b}$ |
| | EPS54 | $4.05 \pm 0.33^{b}$ | $24.6 \pm 1.71^{b}$ | $16.3 \pm 1.07^{a}$ |
| | MTPNSB | $4.02 \pm 0.51^{b}$ | $28.0 \pm 1.85^{a}$ | $16.5 \pm 0.59^{a}$ |
| F (A) | | * | * | * |
| F (B) | | * | * | * |
| F (A × B) | | * | * | * |
| CV (%) | | 9.40 | 6.54 | 5.07 |

| Factors | | Concentrations (%) | | | | | | | |
| --- | --- | --- | --- | --- | --- | --- | --- | --- | --- |
| | | N | | P | | K | | Na | |
| | | Stem, leaves | Seeds | Stem, leaves | Seeds | Stem, leaves | Seeds | Stem, leaves | Seeds |
| The water salinity (A) (‰) | 0 | $1.168 \pm 0.15^{ab}$ | $1.883 \pm 0.21$ | $0.200 \pm 0.02$ | $0.297 \pm 0.012$ | $1.670 \pm 0.095$ | $0.351 \pm 0.037^{a}$ | $0.438 \pm 0.040^{d}$ | $0.032 \pm 0.004^{c}$ |
| | 2 | $1.220 \pm 0.09^{a}$ | $1.932 \pm 0.20$ | $0.192 \pm 0.03$ | $0.293 \pm 0.014$ | $1.650 \pm 0.107$ | $0.300 \pm 0.031^{b}$ | $0.489 \pm 0.064^{c}$ | $0.034 \pm 0.004^{c}$ |
| | 3 | $1.081 \pm 0.08^{c}$ | $1.898 \pm 0.10$ | $0.195 \pm 0.02$ | $0.304 \pm 0.007$ | $1.655 \pm 0.132$ | $0.308 \pm 0.027^{b}$ | $0.536 \pm 0.028^{b}$ | $0.040 \pm 0.005^{b}$ |
| | 4 | $1.109 \pm 0.10^{bc}$ | $1.902 \pm 0.17$ | $0.200 \pm 0.03$ | $0.302 \pm 0.014$ | $1.585 \pm 0.090$ | $0.300 \pm 0.027^{b}$ | $0.589 \pm 0.029^{a}$ | $0.047 \pm 0.006^{a}$ |
| PNSB (B) (1.812 × 10$^{5}$ CFU g$^{-1}$ dry soil) | NAPNSB | $1.087 \pm 0.07^{c}$ | $1.774 \pm 0.06^{b}$ | $0.164 \pm 0.01^{b}$ | $0.290 \pm 0.013^{b}$ | $1.278 \pm 0.127^{c}$ | $0.289 \pm 0.028^{c}$ | $0.630 \pm 0.046^{a}$ | $0.050 \pm 0.002^{a}$ |
| | EPS18 | $1.095 \pm 0.09^{bc}$ | $1.862 \pm 0.15^{ab}$ | $0.200 \pm 0.02^{a}$ | $0.304 \pm 0.009^{a}$ | $1.782 \pm 0.054^{a}$ | $0.317 \pm 0.035^{b}$ | $0.470 \pm 0.038^{c}$ | $0.034 \pm 0.006^{c}$ |
| | EPS37 | $1.157 \pm 0.12^{abc}$ | $2.010 \pm 0.28^{a}$ | $0.207 \pm 0.02^{a}$ | $0.301 \pm 0.012^{a}$ | $1.664 \pm 0.117^{b}$ | $0.284 \pm 0.031^{c}$ | $0.510 \pm 0.032^{b}$ | $0.036 \pm 0.005^{bc}$ |
| | EPS54 | $1.204 \pm 0.14^{a}$ | $1.916 \pm 0.25^{ab}$ | $0.201 \pm 0.03^{a}$ | $0.305 \pm 0.009^{a}$ | $1.730 \pm 0.108^{ab}$ | $0.317 \pm 0.028^{b}$ | $0.532 \pm 0.038^{b}$ | $0.040 \pm 0.005^{b}$ |
| | MTPNSB | $1.180 \pm 0.10^{ab}$ | $1.961 \pm 0.11^{a}$ | $0.211 \pm 0.02^{a}$ | $0.300 \pm 0.017^{a}$ | $1.741 \pm 0.124^{ab}$ | $0.350 \pm 0.031^{a}$ | $0.433 \pm 0.047^{d}$ | $0.035 \pm 0.005^{bc}$ |
| F (A) | | * | ns | ns | ns | ns | * | * | * |
| F (B) | | * | * | * | * | * | * | * | * |
| F (A × B) | | * | ns | ns | ns | * | * | * | ns |
| CV (%) | | 10.3 | 11.7 | 13.0 | 4.61 | 6.95 | 10.6 | 8.68 | 13.5 |

Note:

In the same column, numbers followed by different letters were different from each other at 5% significance (*), ns, not significant; NAPNSB, no application of purple non-sulfur bacteria; EPS18, application of EPS18 strain; EPS37, application of EPS37 strain; EPS54, application of EPS54 strain; MTPNSB, application of the mixture of strains EPS18, EPS37 and EPS54.

treatments applied with a single strain of EPS54 or the PNSB mixture had the N content in straw of 1.204–1.180%, greater than that in the treatment with no bacteria. Nevertheless, the N content in seeds changed insignificantly ($p = 0.915$) under the influence of saline water and had an average value of 0.299%. Moreover, only the treatments supplied with an individual strain of EPS37 or the PNSB mixture possessed greater N content in seeds than that in the treatment with no bacteria.

*P content*

The P concentrations in straw and in seeds did not vary significantly and had a value of 0.197% and 0.299%, respectively, under the influence of irrigating saline water. For the bacterial application, supplying with the PNSB had a better P concentration than the treatment without bacteria. That is, the P contents were 0.200–0.211 > 0.164% in straw, and 0.300–0.305 > 0.290% in seeds, respectively (Table 3).

*K content*

The K concentrations in straw were not significantly different ($p = 0.122$) among the saline levels of the water and had the mean at 0.164%. However, the K content in seeds decreased when the salinity of water was 2–4‰. Moreover, supplying either a single strain of EPS18, EPS37, and EPS54 or their mixture provided a greater K content in straw than that in the treatment without bacteria, with 1.664–1.782 compared with 1.278%, respectively. In seeds, only in the treatments except for the treatment with a single strain of EPS37, supplying the PNSB increased the K content in comparison with the control treatment (Table 3).

*Na content*

Irrigating saline water at 2‰ increased the Na concentration in straw. However, at 3‰ salinity, the Na content went up in seeds. Meanwhile, supplying a single strain of EPS18, EPS37, and EPS54 or their mixture had greater Na content in the treatment without bacteria for both rice stovers (Table 3).

### N, P, K, and Na uptake

*N uptake*

Irrigating 3–4‰ saline water restricted N uptake in straw and in seeds, in comparison with the case using saline water at 0–2‰. On the other hand, in the treatment supplied with the PNSB mixture, the N uptake in straw was the highest with a value of 331.4 mg N pot$^{-1}$. Then, the lowest one was in the no-bacteria case (245.0 mg N pot$^{-1}$). Similarly, the N uptake in seeds was 324.5 ~ 313.0 ~ 314.3 > 284.8 > 213.4 mg N pot$^{-1}$ in the order of supplying the PNSB mixture, a single strain of EPS54, EPS37, and EPS18, and the negative control, respectively (Table 4). When irrigating the saline water at 3‰ and 4‰, the total N uptake decreased, in comparison with that in the treatment applied with no saline water. In addition, supplying either a single strain of EPS18, EPS37, and EPS54 or their mixture resulted in total N uptake of 536.4–656.0 mg N pot$^{-1}$, while in the treatment without bacteria, the result was 458.3 mg N pot$^{-1}$. Applying only the strain of EPS54 or the PNSB mixture along with 2‰ saline water resulted in greater N uptake than the treatment without both bacteria and saline water. Additionally, either individual or mixed three PNSB strains along with with 3–4‰ saline water application all had equivalent total N uptake to that in the treatment with no application from both saline water and PNSB (Table 4).

**Table 4 The influence of the water salinity and the supplementation of exopolymeric subtances providing purple non-sulfur bacteria on the uptake and total uptake of N, P, K and Na in stem, leaves and in seeds of rice cultivated on the salt-contaminated Hong Dan-Bac Lieu soil in greenhouse conditions.**

| Factors | | N uptake | | P uptake | | K uptake | | Na uptake | |
|---|---|---|---|---|---|---|---|---|---|
| | | Stem, leaves | Seeds | Stem, leaves | Seeds | Stem, leaves | Seeds | Stem, leaves | Seeds |
| | | (mg pot$^{-1}$) | | | | | | | |
| The water salinity (A) (‰) | 0 | 316.6 ± 47.5[a] | 323.5 ± 31.9[a] | 54.5 ± 4.1[a] | 51.2 ± 3.1[a] | 457.0 ± 35.0[a] | 61.0 ± 6.8[a] | 117.0 ± 13.0[b] | 5.46 ± 0.81[b] |
| | 2 | 306.5 ± 31.2[a] | 302.0 ± 35.3[a] | 48.2 ± 6.9[b] | 45.5 ± 2.4[b] | 414.0 ± 37.9[b] | 45.1 ± 4.4[b] | 124.0 ± 17.5[ab] | 5.20 ± 0.72[b] |
| | 3 | 253.3 ± 32.5[b] | 274.4 ± 24.9[b] | 45.6 ± 5.4[b] | 44.0 ± 2.9[b] | 386.4 ± 36.0[c] | 44.8 ± 5.3[b] | 124.2 ± 11.1[ab] | 5.72 ± 0.67[b] |
| | 4 | 248.0 ± 26.3[b] | 260.1 ± 29.1[b] | 44.6 ± 7.2[b] | 41.1 ± 2.9[c] | 353.9 ± 33.1[d] | 40.5 ± 4.3[c] | 130.4 ± 12.0[a] | 6.20 ± 0.94[a] |
| PNSB (B)(1.812 × 10$^5$ CFU g$^{-1}$ dry soil) | NAPNSB | 245.0 ± 29.5[c] | 213.4 ± 10.9[c] | 36.9 ± 4.0[c] | 34.2 ± 2.1[d] | 287.2 ± 39.1[c] | 34.7 ± 4.0[d] | 141.4 ± 16.7[a] | 5.77 ± 0.43[ab] |
| | EPS18 | 252.0 ± 27.2[c] | 284.8 ± 30.1[b] | 45.6 ± 5.1[b] | 46.6 ± 3.0[c] | 408.3 ± 23.9[b] | 48.7 ± 6.8[b] | 105.7 ± 12.0[c] | 5.20 ± 0.84[b] |
| | EPS37 | 278.0 ± 29.5[b] | 314.3 ± 49.4[a] | 49.8 ± 7.0[b] | 47.2 ± 2.5[bc] | 399.7 ± 37.3[b] | 44.6 ± 5.1[c] | 122.5 ± 9.2[b] | 5.56 ± 0.88[b] |
| | EPS54 | 299.0 ± 45.4[b] | 313.0 ± 40.7[a] | 49.5 ± 7.3[b] | 50.0 ± 4.1[a] | 428.3 ± 41.9[b] | 52.2 ± 5.0[b] | 129.4 ± 13.8[b] | 6.22 ± 0.98[a] |
| | MTPNSB | 331.4 ± 40.3[a] | 324.5 ± 20.5[a] | 59.1 ± 6.1[a] | 49.4 ± 2.4[ab] | 490.5 ± 35.3[a] | 59.0 ± 5.0[a] | 120.2 ± 15.3[b] | 5.78 ± 0.79[ab] |
| F (A) | | * | * | * | * | * | * | * | * |
| F (B) | | * | * | * | * | * | * | * | * |
| F (A × B) | | * | * | ns | * | * | * | * | ns |
| CV (%) | | 13.1 | 13.3 | 13.5 | 6.74 | 9.57 | 12.0 | 11.8 | 15.4 |

| Factors | | Total uptake | | | |
|---|---|---|---|---|---|
| | | N | P | K | Na |
| | | (mg pot$^{-1}$) | | | |
| The water salinity (A) (‰) | 0 | 640.2 ± 68.1[a] | 105.7 ± 5.7[a] | 518.0 ± 35.1[a] | 122.4 ± 13.0[b] |
| | 2 | 608.4 ± 63.4[a] | 93.8 ± 7.5[b] | 459.1 ± 37.8[b] | 129.1 ± 17.6[ab] |
| | 3 | 527.7 ± 40.4[b] | 89.6 ± 6.6[bc] | 459.1 ± 37.8[b] | 130.0 ± 11.1[ab] |
| | 4 | 507.7 ± 39.2[b] | 85.8 ± 8.1[c] | 394.5 ± 33.8[d] | 136.8 ± 12.6[a] |
| PNSB (B) (1.812 × 10$^5$ CFU g$^{-1}$ dry soil) | NAPNSB | 458.3 ± 35.3[d] | 71.2 ± 5.1[d] | 322.1 ± 40.1[d] | 147.2 ± 16.9[a] |
| | EPS18 | 536.4 ± 52.2[c] | 92.3 ± 6.6[c] | 457.0 ± 25.0[bc] | 111.0 ± 12.0[c] |
| | EPS37 | 592.3 ± 66.5[b] | 97.1 ± 8.3[bc] | 444.3 ± 35.4[c] | 128.1 ± 9.6[b] |
| | EPS54 | 611.8 ± 60.9[b] | 99.5 ± 8.4[b] | 480.6 ± 42.8[b] | 135.7 ± 14.2[b] |
| | MTPNSB | 656.0 ± 48.9[a] | 108.5 ± 6.3[a] | 549.4 ± 33.6[a] | 126.0 ± 15.2[b] |
| F (A) | | * | * | * | * |
| F (B) | | * | * | * | * |
| F (A * B) | | ns | ns | * | * |
| CV (%) | | 10.5 | 8.07 | 8.54 | 11.4 |

**Note:**
In the same column, numbers followed by different letters were different from each other at 5% significance (*), ns, not significant; NAPNSB, no application of purple non-sulfur bacteria; EPS18, application of EPS18 strain; EPS37, application of EPS37 strain; EPS54, application of EPS54 strain; MTPNSB, application of the mixture of strains EPS18, EPS37 and EPS54.

*P uptake*

The P uptake in straw and in seeds peaked in the treatment with 0‰ salinity (54.5 and 51.2 mg P pot$^{-1}$, respectively). Then, in the treatments applied with 2–3‰ saline water, it was 45.6–48.2 and 44.0–45.5 mg P pot$^{-1}$, and the lowest P uptake was recorded in the

treatment applied with 4‰ saline water (44.6 and 41.1 mg P pot$^{-1}$, respectively). In addition, in the treatments supplied with a single strain of EPS18, EPS37, and EPS54 or their mixture, the P uptake was greater than that in the treatment without bacteria (Table 4). The total P uptake in the treatment without saline water was 105.7 mg P pot$^{-1}$, greater than that in the treatments irrigated with 2‰ (93.8 mg P pot$^{-1}$), 3‰ (89.6 mg P pot$^{-1}$), and 4‰ (85.8 mg P pot$^{-1}$) saline water ($p < 0.00009$). Additionally, the supplementation of the mixture of EPS18, EPS37, and EPS54 had the highest total P uptake. Furthermore, in the treatments supplied with a single strain of EPS18, EPS37, and EPS54, the total P uptake was 92.3, 97.1, and 99.4 mg P pot$^{-1}$, respectively, which were all greater than that in the treatment without bacteria (Table 4).

*K uptake*

Along with the increase in salinity of the water from 0‰ to 4‰, the K uptake was 457.0 > 414.0 > 386.4 > 353.9 mg K pot$^{-1}$ in straw and 61.0 > 45.1 > 44.8 > 40.5 mg K pot$^{-1}$ in seeds. Moreover, the supplementation of the PNSB mixture had K uptake in straw and in seeds of 490.5 and 59.0 mg K pot$^{-1}$, respectively, which was greater than those in the treatments supplied with a single strain of EPS18, EPS37, and EPS54, whose values were 399.7–428.3 and 44.6–52.2 mg K pot$^{-1}$ ($p = 0.00007$). Meanwhile, the lowest K uptake belonged to the treatment without bacteria, with 287.2 and 34.7 mg K pot$^{-1}$, respectively (Table 4). Increasing the concentrations of salt in the water from 0‰, 2‰, 3‰ to 4‰ resulted in a reduction in the total K uptake as well. Furthermore, supplying the mixture of the PNSB resulted in the highest total K uptake (549.4 mg K pot$^{-1}$). Then, the lowest result was observed with the treatment without bacteria (322.1 mg K pot$^{-1}$). The interaction of analytic results revealed that, in the same saline levels, both the single and the mixed strains of PNSB contributed to the increase in the total K uptake, in comparison with the treatment without bacteria application (Table 4). Moreover, the highest total K uptake was in the treatment with the PNSB mixture at 0‰ salinity and different significantly from other treatments, while the lowest one was in the treatments without PNSB (Table S6). However, in the treatments with the PNSB mixture, the greater salinity was, the lower total K uptake became.

*Na uptake*

At 4‰ of water salinity, the Na uptake increased in straw and in seeds, in comparison with that in the no saline water case. In addition, the individual use of PNSB EPS18, EPS37, and EPS54 or the application of the mixture of the three PNSB strains had lower Na content accumulated in straw than that in the treatment without bacteria, with 105.7–129.4 compared with 141.4 mg Na pot$^{-1}$. Nevertheless, both the single and the mixed application of the three PNSB strains did not reduce the Na uptake in seeds (Table 4). The treatments applied with 4‰ saline water had the total Na uptake at 136.8 mg Na pot$^{-1}$, remarkably greater than those in the treatments with no saline water applied (122.4 mg Na pot$^{-1}$). Moreover, in the treatments with the application of EPS18, EPS37, and EPS54 individually or mixed, total Na uptake fluctuated in the range of 110.9–135.7 mg Na pot$^{-1}$, while in the treatment without bacteria, the total Na uptake went up to 147.2 mg Na pot$^{-1}$. The analysis
of interaction revealed that, at 2–3‰ salinity of the water, supplying the single strain of EPS18 and EPS37 or the PNSB mixture had a lower total Na uptake than that in the treatment without bacteria (Table 4). The treatment with the PNSB mixture at 0‰ salinity reduced the total Na uptake in comparison with the treatment without bacteria at 2–4‰ salinity (Table S7). Moreover, from 2–3‰, the treatment with bacteria decreased the total Na uptake, except for the treatment with the single EPS54 strain at 2‰ salinity. However, at 0 and 4‰ salinity, the influence of the PNSB on the total uptake was not significant.

## The influence of the PNSB and the saline irrigation on the rice growth and yield

### Rice growth

Rice plant height changed under the influence of both factors. In detail, between 0‰ and 2‰ salinity, plant heights were equivalently 82.4 and 81.0 cm, respectively. However, in the treatments applied with 3–4‰ saline water, the height dropped significantly (79.5 and 78.8 cm) as compared with the 0‰ salinity case. In addition, in the treatments supplied with a single strain of EPS18, EPS37, and EPS54 or their mixture, plant height was 79.0–83.3 cm, while in the treatment without bacteria, the number was only 76.5 cm. At 2–4‰ salinity, the panicles were shortened, in comparison with those in the treatment without saline water irrigated, with 19.0–19.4 cm compared to 20.1 cm, respectively. Nonetheless, the treatments applied with a single strain of EPS37 and EPS54 or the mixture of EPS18, EPS37, and EPS54 possessed panicles that were 19.5–20.3 cm long and longer than those in the treatment without bacteria (18.4 cm) (Table 5).

### Rice yield components

The number of panicles on a pot and the percentage of filled seeds decreased when being applied with water from 2‰ and above saline levels, while the number of seeds per panicle started to drop at 3–4‰ salinity. Besides that, the treatments supplied with a single strain of EPS18, EPS37, and EPS54 or their mixture had a greater number of panicles per pot (16.3–18.0 panicles) and a greater number of seeds per panicle (88.4–92.0 seeds) in comparison with those in the control treatment, whose result was 15.0 panicles and 79.9 seeds, respectively. However, for the percentage of filled seeds, except for the treatment with the single EPS37 strains, supplying the PNSB had greater results than the treatment without bacteria. The panicle length was also affected by both factors. In detail, irrigating the saline water resulted in a drop in the length of rice panicles. The supplementation of the PNSB, except for the EPS18 strain alone, resulted in longer panicles, in comparison with that treated without bacteria, with 19.6–20.3 compared with 18.4 cm. However, the 1,000-seed weight varied insignificantly among treatments ($p > 0.05$) (Table 5).

### Rice grain yield

The grain yield in the treatments irrigated with saline water was different at 5% significance ($p = 0.00001$) from that in the treatment without applied saline water. To be more specific, grain yield in the treatments applied with 0‰ and 2‰ saline water was 22.5 and 21.3 g pot$^{-1}$, respectively, and greater than those in the treatments applied with 3‰ and 4‰ saline water, with 19.6 and 17.5 g pot$^{-1}$, respectively. Meanwhile, the treatments

**Table 5 The influence of the water salinity and the supplementation of exopolymeric subtances providing purple non-sulfur bacteria on the growth, the yield components and the grain yield of rice plants cultivated on the salt-contaminated Hong Dan-Bac Lieu soil in greenhouse conditions.**

| Factors | | Plant height (cm) | Panicle length (cm) | Panicles number pot⁻¹ (panicles) | Seeds number panicle⁻¹ (Seeds) | Filled seeds ratio (%) | 1,000-seed weight (g) | Grain yield (g pot⁻¹) |
|---|---|---|---|---|---|---|---|---|
| The water salinity(A) (‰) | 0 | 82.4 ± 2.6ᵃ | 20.1 ± 0.6ᵃ | 18.1 ± 1.1ᵃ | 91.0 ± 3.8ᵃ | 81.9 ± 6.2ᵃ | 22.4 ± 1.1 | 22.5 ± 2.5ᵃ |
| | 2 | 81.0 ± 1.9ᵃᵇ | 19.4 ± 1.2ᵇ | 17.0 ± 1.1ᵇ | 89.0 ± 5.2ᵃᵇ | 75.2 ± 5.3ᵇ | 23.2 ± 1.7 | 21.3 ± 1.3ᵃ |
| | 3 | 79.5 ± 2.5ᵇᶜ | 19.3 ± 1.3ᵇ | 16.3 ± 1.0ᵇ | 87.3 ± 4.4ᵇᶜ | 68.0 ± 5.8ᵇ | 23.2 ± 2.0 | 19.6 ± 0.8ᵇ |
| | 4 | 78.8 ± 2.1ᶜ | 19.0 ± 0.8ᵇ | 15.1 ± 1.4ᶜ | 86.0 ± 2.3ᶜ | 59.8 ± 5.3ᶜ | 23.0 ± 0.6 | 17.5 ± 1.3ᶜ |
| PNSB (B) (1.812 × 10⁵ CFU g⁻¹ dry soil) | NAPNSB | 76.5 ± 1.2ᵈ | 18.4 ± 0.4ᶜ | 15.0 ± 0.9ᶜ | 79.9 ± 2.2ᶜ | 65.2 ± 5.3ᵈ | 23.6 ± 0.7 | 17.7 ± 1.2ᵈ |
| | EPS18 | 79.0 ± 1.9ᶜ | 19.2 ± 0.9ᵇᶜ | 18.0 ± 1.3ᵃ | 90.1 ± 4.2ᵃᵇ | 71.3 ± 4.7ᵇᶜ | 23.0 ± 0.9 | 20.0 ± 1.0ᵇᶜ |
| | EPS37 | 82.2 ± 3.6ᵃᵇ | 19.6 ± 0.8ᵃᵇ | 16.3 ± 1.0ᵇ | 89.9 ± 5.0ᵃᵇ | 68.1 ± 6.2ᶜᵈ | 23.3 ± 2.3 | 19.7 ± 1.4ᶜ |
| | EPS54 | 81.1 ± 2.5ᵇ | 20.3 ± 1.3ᵃ | 16.3 ± 1.1ᵇ | 88.4 ± 3.4ᵇ | 75.1 ± 6.9ᵃᵇ | 22.4 ± 1.4 | 21.2 ± 2.9ᵃᵇ |
| | MTPNSB | 83.3 ± 2.2ᵃ | 19.5 ± 1.5ᵇ | 17.6 ± 1.5ᵃ | 92.0 ± 4.8ᵃ | 76.6 ± 5.2ᵃ | 22.6 ± 1.5 | 22.6 ± 1.0ᵃ |
| F (A) | | * | * | * | * | * | ns | * |
| F (B) | | * | * | * | * | * | ns | * |
| F (A × B) | | ns | ns | ns | ns | ns | ns | ns |
| CV (%) | | 3.24 | 6.00 | 7.38 | 4.90 | 8.60 | 8.00 | 9.99 |

**Note:**
In the same column, numbers followed by different letters were different from each other at 5% significance (*), ns, not significant; NAPNSB, no application of purple non-sulfur bacteria; EPS18, application of EPS18 strain; EPS37, application of EPS37 strain; EPS54, application of EPS54 strain; MTPNSB, application of the mixture of strains EPS18, EPS37 and EPS54.

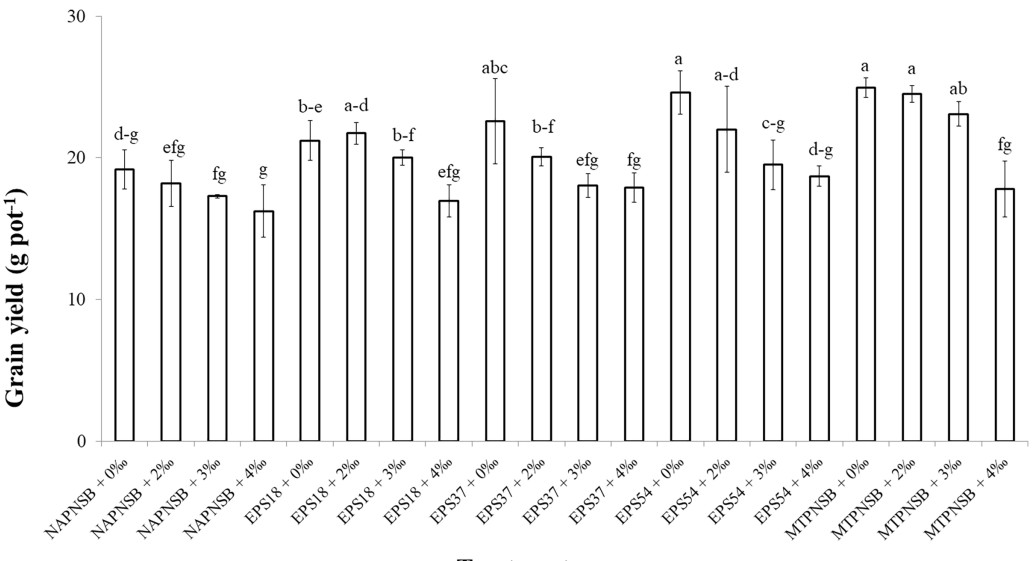

**Figure 1 The interaction between the water salinity and the supplementation of the exopolymeric substances providing purple non-sulfur bacteria on rice yield on salt-contaminated Hong Dan-Bac Lieu soil in greenhouse condition.** NAPNSB, no application of purple non-sulfur bacteria; EPS18, application of EPS18 strain; EPS37, application of EPS37 strain; EPS54, application of EPS54 strain; MTPNSB, application of the mixture of strains EPS18, EPS37 and EPS54.

applied with bacteria had significantly different grain yields from those without bacteria. In overall, the treatments supplied with the PNSB mixture had a greater grain yield (22.6 g pot$^{-1}$) than the treatments supplied with an individual strain of EPS18 and EPS37 (20.0 and 19.7 g pot$^{-1}$). The lowest grain yield was found in the treatments without bacteria (17.7 g pot$^{-1}$) (Table 5). In the case of no saline water irrigated, the application of a single strain of EPS37 and EPS54 or the PNSB mixture resulted in a greater grain yield than in the control treatment. Moreover, at 2‰ salinity, supplying the PNSB individually or in mixture provided a greater grain yield than the treatment without bacteria. However, at 3‰ salinity, only the treatment applied with the PNSB mixture had a greater grain yield (23.1 g pot$^{-1}$) than the control treatment (17.3 g pot$^{-1}$). Furthermore, when the salinity reached 4‰, treatments with or without the PNSB had statistically the same grain yield values (Fig. 1).

## DISCUSSION

### The *L. sphaeroides* improved the properties of the acidic salt-affected soil

Acidity and salinity are considered major obstacles to rice cultivated on salt-affected acid-sulfate soil (*Huang et al., 2017*; *Syed et al., 2021*). Moreover, low pH reduces the availability of macronutrients but increases that of toxins, resulting in changes in plants' absorption of nutrients (*Tsai & Schmidt, 2021*). Thus, acidity adversely affects the growth and the capability of accumulating nutrients of the plants (*Huang et al., 2017*; *Tsai & Schmidt, 2021*). Furthermore, high Na$^+$ concentration also interferes with the process of taking nutrients and the growth of rice (*Irakoze et al., 2021*; *Syed et al., 2021*), while rice is one of the major crops in Vietnam, whose soil has been damaged by salinity and acidity. In the current study, the soil salinity was positively correlated with the saline levels in the water applied for rice cultivation, shown *via* the EC value and the amount of exchanging Na$^+$ in soil. Therefore, an approach to ameliorate the pH, and to decease the EC and the Na$^+$ concentration in paddy soil is required. The results in the current study indicated that supplying a single strain of EPS18 and EPS37 or the mixture of the three PNSB EPS18, EPS37, and EPS54 improved pH$_{H2O}$, and reduced the EC and the Na$^+$ concentration, with 3.88–3.99 compared to 3.75, with 0.68–0.82 compared to 0.96 mS cm$^{-1}$, and 1.144% compared to 0.769% in the treatment without bacteria, respectively (Table 2). This is because these bacteria belonged to PNSB which has been proven to be able to improve soil functions by increasing carbon and nitrogen contents, soil aggregation and stability and soil water status in adverse environments (*Mager & Thomas, 2011*; *Chamizo et al., 2012*). In the meantime, this bacteria group can produce EPS, which is considered to be a compound that can cut off the salinity in soil *via* functional groups within (*Nunkaew et al., 2015b*; *Panwichian et al., 2011*). According to *Panwichian et al. (2011)*, the PNSB are capable of reducing Na$^+$ content by secreting EPS to immobilize Na$^+$ or to accumulate Na$^+$ in bacterial biomass. The correlation analysis of the current study indicated the interaction between the bacterial density in soil and the soil Na$^+$ content (r = 0.7635) (Fig. 2A). This should show that the appearance of the PNSB changed the Na$^+$ concentration in soil.

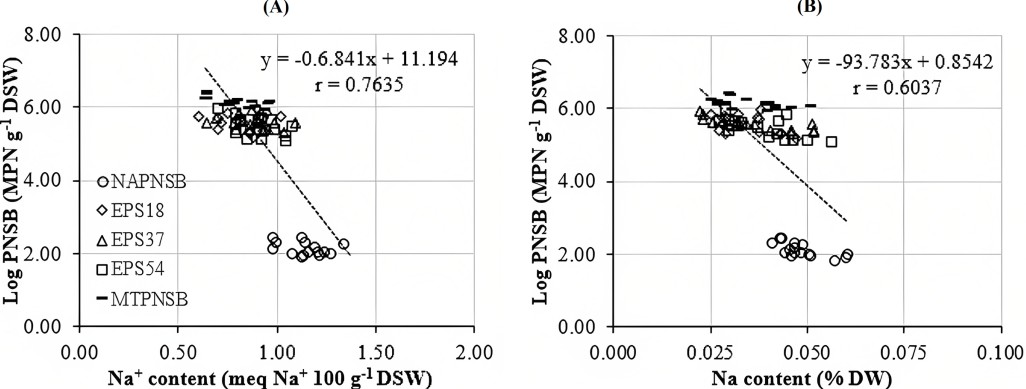

**Figure 2 Correlation between PNSB and (A) soil Na concentration; (B) and plant Na concentration.** NAPNSB, no application of purple non-sulfur bacteria; EPS18, application of EPS18 strain; EPS37, application of EPS37 strain; EPS54, application of EPS54 strain; MTPNSB, application of the mixture of strains EPS18, EPS37 and EPS54.

Other bacteria, such as *Alcaligenes* sp., are found to be able to reduce saline stress on rice plants by producing EPS (*Fatima et al., 2020*). In the current study, the ability to produce EPS by the strains of EPS18, EPS37, and EPS54 was almost the same, with 1.86, 1.97, and 1.82 mg L$^{-1}$ in microaerobic light and 1.14, 1.59, and 1.24 mg L$^{-1}$ in aerobic dark conditions (our preliminary study, LNT Xuan, ND Trong, VM Thuan, TC Nhan, LT Quang, LTM Thu, NHM Anh, NQ Khuong, 2023, Unpublished data). However, the capability of reducing Na$^+$ content of these bacterial strains was insignificantly different. For further evaluation, the more saline it is, the more EPS the PNSB produce (*Zeng et al., 2016*; *Ji et al., 2020*). The bacteria, *Lactobacillus plantarum* VAL6, have been found to produce exopolysaccharide 6.4 times greater in stress caused by low pH (pH 3.0). However, being stressed by low pH and high Na$^+$, the *L. plantarum* VAL6 produces exopolysaccharides with charged components (*Nguyen et al., 2021*). Therefore, these strains should be further evaluated under different stresses for EPS production.

Apart from that function, there are two types of EPS in the soil produced by bacteria. The first one is the loosely bound EPS, which is less condensed, and easily released into the environment; the other one is the tightly bound EPS, which is a more condensed, and firmly bonds to the cells and soil particles (*Rossi & De Philippis, 2015*; *Rossi, Mugnai & De Philippis, 2018*). The EPS matrix performs as a repository for nutrients and metabolites produced by microorganism (*Swenson et al., 2018*), and cations (*De Philippis, Colica & Micheletti, 2011*; *Rossi & De Philippis, 2015*), it can also hold water, and increase water uptake and retention in soils (*Colica et al., 2014*; *Adessi et al., 2018*). Thereby the matrix greatly enhances the nutrient contents in the soil. This contributed to explaining why nutrient contents in the soil in treatments with the bacteria were greater than those without bacteria. Consequently, these strains indirectly contributed to the improvement in the availability of macronutrients and limited the harm from soil toxins. This helped in interpreting why the amount of available N, P, and K increased in the treatments supplied with bacteria. Previous studies have also revealed that the bacteria of *R. palustris* possessing abilities to fix N and solubilize P and K were applied in order to enhance plant growth

(*Khuong et al., 2018*, *2021*). Similarly, the PNSB strains EPS18, EPS37, and EPS54 were found to be capable of fixing N and solubilizing P and K. As in the outcome, their application increased the concentrations of the $NH_4^+$ from 2.1 to 3.0 mg $NH_4^+$ $kg^{-1}$, soluble P from 4.2 to 15.5 mg P $kg^{-1}$, and exchangable K from 0.013 to 0.026 meq $K^+$ 100 $g^{-1}$, in comparison with those in the treatments without bacteria (Table 2). This can be interpreted that the PNSB can fix N, due to the genes they have, including *nifH*, *vnfG*, and *anfG* (*Sakpirom et al., 2017*), and solubilize insoluble P forms in soil because of their ability to secrete acid phosphatase enzyme and siderophores (*Khuong et al., 2018, 2020*). Previous studies indicated that the PNSB belonging to *Rhodopseudomonas* and *Luteovulum* genera are capable of solubilizing P in saline and tolerating acidic conditions, and aimed to ameliorate the availability of P in soil (*Khuong et al., 2018, 2021*). This can be seen in the reduction in the amount of insoluble P compounds in soil in the treatments supplied with the mixture of EPS18, EPS37, and EPS54, whose concentrations of Al-P, Fe-P, and Ca-P valued at 59.4, 177.4, and 48.9 mg P $kg^{-1}$, respectively, and lower than those in the treatments without bacteria, with 67.5, 200.7, and 69.4 mg P $kg^{-1}$, respectively. However, the P solubilization was different from each strain. In particular, the strain of EPS18 and EPS37 had a lower concentration of Al-P than the control treatment, while applying the strain EPS54 had an equivalent amount of Al-P, compared to the control treatment. Nevertheless, supplying the PNSB strain EPS18, EPS37, or EPS54 did not reduce the Fe-P significantly, while the combination of them made a drop in the concentration of Fe-P in soil. Moreover, using the three strains EPS18, EPS37, and EPS54 separately or in mixture all had a Ca-P concentration of 48.9–57.5 mg P $kg^{-1}$, lower than those in the control treatment (Table 2). Thus, the improvements in the nutrient availability and limitation in Na element by the PNSB application, the performance of rice under saline condition Therefore, the N fixation and P solubilization of these strains should be further specifically investigated in the future studies.

## The *L. sphaeroides* improved the tolerance of rice under acidic saline condition

*Abdelaziz et al. (2018)* assumed that proline is synthesized to maintain cellular osmosis and to protect the structure of enzymes, proteins, membrane, and the cell itself in dry condition; that is, proline helps plants to tolerate stress. Rice plants accumulate greater proline concentration in a saline stress condition than in a normal one to adjust the osmosis, enhance water absorption, restrict $Na^+$ and $Cl^-$ uptake, and transport from roots to stem (*Wu et al., 2003*; *Bhusan et al., 2016*). Consistently, the proline concentration in the treatments applied with 2–4‰ saline water was 3.96–5.28 µmol $g^{-1}$ DW, while in the treatment without saline water, the amount of proline was 3.64 µmol $g^{-1}$ DW. It can be observed that the increase in the salinity of the water resulted in the increase in the accumulation of proline in plants. Therefore, the amount of proline produced is an expression of stress in rice plants. Interestingly, in the treatments without bacteria, supplying an individual strain of EPS18, EPS37, and EPS54 or their mixture, the proline concentration was 5.60 > 4.32 ~ 4.18 ~ 4.05 ~ 4.02 µmol $g^{-1}$ DW (Table 3). This is in consistent with the experiment where the $Na^+$ concentration in soil was also reduced by

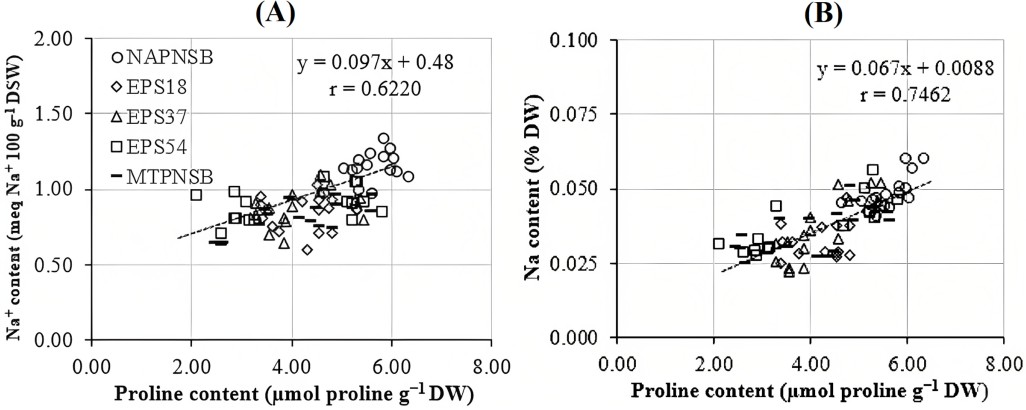

**Figure 3 Correlations (A) between soil Na$^+$ concentration and leaf, stem proline content; and (B) between plant Na concentration and leaf, stem proline content.** NAPNSB, no application of purple non-sulfur bacteria; EPS18, application of EPS18 strain; EPS37, application of EPS37 strain; EPS54, application of EPS54 strain; MTPNSB, application of the mixture of strains EPS18, EPS37 and EPS54.

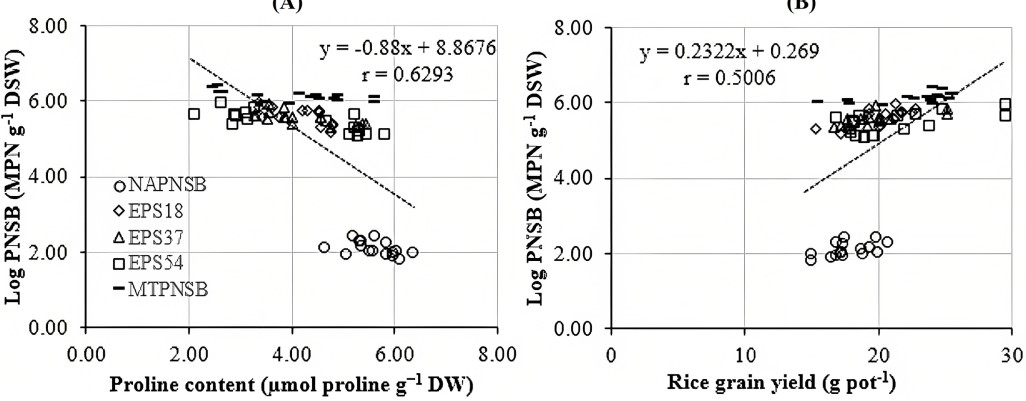

**Figure 4 Correlations between PNSB soil population and (A) leaf, stem proline content and (B) grain yield.** NAPNSB, no application of purple non-sulfur bacteria; EPS18, application of EPS18 strain; EPS37, application of EPS37 strain; EPS54, application of EPS54 strain; MTPNSB, application of the mixture of strains EPS18, EPS37 and EPS54.

the application of the PNSB. In other words, it can be interpreted that the application of the PNSB immobilized the exchanging Na$^+$ amount in soil, leading to rice that was less influenced by saline stress and produced less proline as well. The result also demonstrated the interaction between the amount of proline produced and the Na$^+$ concentration in soil (r = 0.6220) and in plants (r = 0.7462) (Figs. 3A and 3B). That is, the greater the Na$^+$ concentration appeared in soil, the greater Na content accumulation in plants, and the greater proline was produced. Figure 4A illustrated that the treatments supplied with bacteria had a low proline concentration, with a correlation coefficient of r = 0.6293, *i.e.*, the bacteria reduced the saline stress in rice plants. Overwise speaking, the PNSB strains EPS18, EPS37, and EPS54 in the current study were the factor that helped rice plants to

overcome saline stress. Other strains of PNSB have been proven for reducing saline stress on rice *via* proline content in plants (*Khuong et al., 2023a*).

## The *L. sphaeroides* improved N, P, K, and Na uptake

The N concentration in straw and the K concentration in seeds decreased when the salinity of the water increased to 0‰, 2‰, 3‰, and 4‰. Meanwhile, the N and P concentrations in seeds and the P and K concentrations in straw remained statistically unchanged among the saline water levels. Nevertheless, the Na concentration in straw and in seeds increased positively to the saline levels of the water. This showed the influence of the soil salinity on the nutrient contents within the plants. However, the supplementation of the mixture of the three PNSB boosted the amount of N, P, and K and reduced the Na concentration in straw and in seeds. This can be inferred that supplying the PNSB strains promoted soil pH, improving the availability of N, P, and K (Table 2). This played a role in increasing N, P, and K uptake and concentrations within the plant. However, the nutrient providing capacity among the strains were different. In particular, supplying the single strain of EPS54 shared the same trend as supplying the mixture of the three PNSB for the amount of N, P, K, and Na, except for N concentration in seeds. Supplying the single strain of EPS37 increased the N and P concentrations in seeds and the P and K concentrations in straw. Meanwhile, it reduced Na content in both stovers. Supplying the single strain of EPS18 increased P and K content and reduced Na content. However, it did not change the N content both in straw and in seeds (Table 3). The bacterial density in soil and the Na content in seeds were closely correlated (r = 0.7635) (Fig. 2B). This demonstrated that the treatments supplied with either a single strain of EPS18, EPS37, and EPS54 or their mixture were effective in decreasing the amount of Na in seeds. The different functions possessed by different strains in the same mixture were also observed in the study by *Khuong et al. (2023a)*. However, the current strains should be further investigated for K solubilization due to the increases in K content and K uptake found in plants. This has been also found in the study by *Khuong et al. (2023b)*.

When the salinity reached 4‰, the N, P, and K uptake dropped and the Na uptake rose both in straw and in seeds. Applying the saline water also lowers the nutrient uptake of plants (*Fageria, Gheyi & Moreira, 2011*). In addition, supplying a single strain of EPS18, EPS37, and EPS54 or their mixture increased the N, P, and K content in straw and in seeds, except for supplying the single strain of EPS18 for N uptake in straw. This can be explained that the bacteria produced EPS limiting the movement of $Na^+$ ion (*Nunkaew et al., 2015b*; *Panwichian et al., 2011*), which inhibited the transportation of $Na^+$ into rice. Bacterial strains of *Bacillus aerius* ATHM 35, *Arthrobacter luteolus* ATHM 36, *Halomonas eurihalina* ATHM 37, *B. tequilensis* UPMRB9, and *B. aryabhattai* UPMRE6 have been isolated from a hypersaline environment containing functional groups, including hydroxyl, alkyne, amides, and carboxyl, in order to bind and chelate $Na^+$ in the soil (*Isfahani et al., 2018*; *Shultana et al., 2020b*). Thus, Na accumulated less within the rice (Table 4). This result was in accordance with the study by *Khuong et al. (2021)*, where the supplementation of *L. sphaeroides* W03 and W11 contributed to reducing Na uptake in rice. Furthermore, *L. sphaeroides* secreted ALA, promoting rice tolerance to overcome

saline stress and their nutrient uptake (*Khuong et al., 2023a*). However, using a single strain of EPS18, EPS37, and EPS54 or their mixture did not reduce the Na content in straw and did not significantly affect the Na content in seeds (Table 3). This could be interpreted that, before flowering, the saline accumulation occurs due to the control of roots and the Na distribution to plant parts, but, after flowering, it is caused by the transpiration in the panicles (*Asch et al., 1999*). Nevertheless, supplying a single strain of EPS18, EPS37, and EPS54 or their mixture followed the opposite trend. That is, the total N, P, and K uptake increased, and the total Na uptake decreased (Table 4).

### The EPS providing *L. sphaeroides* improved the growth, and the yield of rice

Irrigating saline water from 2‰ above, the panicle length, the number of panicles per pot, and the percentage of filled grains remarkably decreased. This resulted in a reduction in rice yield by 12.9–22.2% in the treatments irrigated with 3–4‰ saline water (Fig. 1). This was in accordance with the study by *Irakoze et al. (2021)*, where rice yield decreased by 50% when being applied with 3‰ saline water. As demonstrated by *Aguilar et al. (2017)*, applying saline water at the vegetative stage of rice lowered plant height, number of panicles, 1,000-seed weight, number of filled seeds, harvest index, and grain yield due to the osmosis and toxicity of $Na^+$ ion (*Mensah et al., 2006*; *Rengasamy & Olsson, 1993*). However, the average 1,000-seed weight was insignificantly different under the influence of both saline water irrigation and the supplementation of the PNSB (Table 5). On the other hand, supplying a single strain of EPS18, EPS37, and EPS54 increased plant height, number of panicles per pot, and number of seeds per panicle. Meanwhile, supplying the mixture of EPS18, EPS37, and EPS54 enhanced agronomic parameters and yield components previously mentioned. In addition, supplying a single strain of EPS18, EPS37, and EPS54 or their mixture did not change the weight of 1,000 seeds (Table 5). Therefore, the application of a single strain of EPS18, EPS37, and EPS54 increased grain yield by 11.3–19.8%. Meanwhile, combining the three PNSB strains altogether increased the grain yield by 27.7% (Fig. 1) due to the positive correlation between the bacterial density and the rice grain yield (Fig. 4B). This was consistent with the study by *Nunkaew et al. (2015a)*, where supplying the bacteria of *R. palustris* TN114 and PP803 stimulated rice growth in saline stress conditions. Moreover, in the current study, since the strains of EPS18, EPS37, and EPS54 excreted EPS, soil salinity was decrease (Table 2) and, thereby, grain yield was enhanced (Fig. 1). *L. sphaeroides* provides EPS, a key compound for metabolic processes, and helps bacteria to colonize plant roots and the rhizosphere, resulting in an improvement of the growth and yield of plants (*Banerjee et al., 2019*). Furthermore, when supplying with EPS providing saline-tolerating bacteria, including *B. tequilensis* UPMRB9 and *B. aryabhattai* UPMRE6, the photosynthesis, transpiration, and stomatal conductance are better, resulting in better plant yield (*Shultana et al., 2020a*). Ultimately, supplying the PNSB strains of EPS18, EPS37, and EPS54 individually or in mixture all improved rice grain yield in saline conditions. Nevertheless, although rice growth and yield benefited from the EPS18, EPS37, and EPS54 strains by their ability to reduce salinity, and to increase nutrient availability, other mechanisms may also involve, such as

pyrroloquinoline quinone (*Lo et al., 2023*), IAA (*Yusof & Hasmoni, 2023*), siderophores (*Nookongbut et al., 2018*), and ALA (*Khuong et al., 2023a*). Therefore, the strains investigated in the current study should be latter on tested for producing the above plant growth promoting substances. Moreover, the PNSB are sensitive and behave differently to different concentrations of nutrients. For instance, in the study by *Shaikh et al. (2023)*, the growth of PNSB is restricted during nutrient deficiency. Moreover, when N is deficient, biofilm formation is enhanced; when Mg is not applied, PNSB produce protein the most; and the least protein content is produced when Ca is deficient. Therefore, an appropriate amount of nutrient fertilizers should be further considered along with a suitable density of bacteria used according to their specific functions. Finally, a field test should be made, when the three strains have been investigated for their specific functions and their response to different nutrient rates, so as to clarify their true performance and their possibility to be commercialized and transferred to farmers.

## CONCLUSIONS

Both factors remarkably affected the features of rice plants grown on acidic salt-affected soil. Irrigating the 2‰ saline water reduced plant height by 2.63 cm, panicle length by 0.90 cm, the number of panicles per pot by 2.00 panicles $pot^{-1}$, the number of seeds per panicle by 2.76 seeds $panicle^{-1}$, the percentage of filled grains by 14.2%, and the rice grain yield by 3.00 g $pot^{-1}$ on acidic salt-affected soil. In the meantime, supplying an individual strain of EPS18, EPS37, and EPS54 or their mixture ameliorated the rice plant height to 83.3 ± 2.2 cm, the length of panicles to 20.3 ± 1.3 cm, the number of panicles per pot to 18.0 ± 1.3 panicles $pot^{-1}$, the seeds number per panicle to 92.0 ± 4.8 seeds $panicle^{-1}$, the percentage of filled grains to 76.6 ± 5.2, and the rice grain yield to 22.6 ± 1.0 g $pot^{-1}$. It also enhanced $pH_{H2O}$, $NH_4^+$, available P, and N, P, and K uptake and reduced Na uptake. This strongly indicated that the *L. sphaeroides* EPS18, EPS37, and EPS54 strains were able to improve the performance of rice plants which were planted under saline condition (soil and water). Those strains should be further applied in the fields where soil is contaminated by salt intrusion in Vietnam and generally in regions damaged by high salinity around the globe. Moreover, the application of these strains should be combined with different concentrations of fertilizers to navigate the most efficient and desired used of the bacteria, and to limit the amount of chemical fertilizers used for ensuring sustainable agriculture.

### Funding
This research was funded by Can Tho University, grant number T2022-88. The funders had no role in study design, data collection and analysis, decision to publish, or preparation of the manuscript.

## Grant Disclosures

The following grant information was disclosed by the authors:
Can Tho University: T2022-88.

## Competing Interests

The authors declare that they have no competing interests.

## Author Contributions

- Nguyen Quoc Khuong conceived and designed the experiments, performed the experiments, analyzed the data, prepared figures and/or tables, authored or reviewed drafts of the article, and approved the final draft.
- Nguyen Minh Nhat performed the experiments, analyzed the data, prepared figures and/or tables, authored or reviewed drafts of the article, and approved the final draft.
- Le Thi My Thu conceived and designed the experiments, performed the experiments, analyzed the data, authored or reviewed drafts of the article, and approved the final draft.
- Le Vinh Thuc conceived and designed the experiments, authored or reviewed drafts of the article, and approved the final draft.

## DNA Deposition

The following information was supplied regarding the deposition of DNA sequences:
The sequences are available at GenBank: OR044031, OR044032 and OR044033.

## Data Availability

The data is available in the Supplemental File.

## Supplemental Information

Supplemental information for this article can be found online at http://dx.doi.org/10.7717/peerj.16943#supplemental-information.

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
