# Peer review of "Influence of purple non-sulfur bacterial augmentation on soil nutrient dynamics and rice (Oryza sativa) growth in acidic saline-stressed environments"

_PeerJ, doi:10.7717/peerj.16943_

## Round 0.1 · original submission · Major Revisions

The reviewers gave very insightful comments to improve the quality of the manuscript. Please address all the questions and comments carefully.

·

Basic reporting

- The title should be shortened.
- The abstract is complex and not clear. Please rewrite.
- Why you selected these PNSB strains why not other? Add more reasons.
- Add what are the important features of PNSB that make it unique than other bacteria in introduction.
- Use percentage (%) symbol properly in line 20, and 26. It implied to whole manuscript.
- Not clear about the statement that “no bacteria is used” and you mentioned different strains.
- English of the manuscript should be improved. It has lot of grammatical and typo errors. For instance, in line 78 “T” should be capitalized in “the”. Please consider this comment for all such types of words.
- Centigrade symbol should be used properly in line 91.
- In the footnote of table 1, EC, and CEC full form should be mentioned.
- Why have you selected the time duration from June 2022 to November 2022 ? Provide valid reason. Line 89-90.
- Percentage symbol in line 85 should be used properly.
- Please add schematic diagram showing the experimental setup.
- Line 99. Check the spelling.
- Percentage symbol error – Line 178.
- I think figure 2b x-axis should be same as 2a.

Experimental design

- The novelty of the study should be clearly defined.
- What are the research questions? Define them clearly.
- Methodology section is very complex and not clear. For instance, you are talking about different things every time. Initially you never mentioned the number of pots. Also it is not clear what you have used as pot. You mentioned that 500 rice seeds were divided into 5 portions and again you mentioned that each pot was sowed with 6 seeds. Please rewrite the complete methodology. Lines 70-169
- You mentioned that you have utilized fertilizer following the recommendation? Which recommendation? Line 101.
- Why 20, 27, 34, 41, and 55 DAS ? Line 107.
- Why 13, 18 and 30 DAS? Line 110.
- How you came to know that PNSB is providing you the certain amount of PNSB? Line 112-116.
- Provide brief description of bacterial densities method. Line 136.
- Why biochemical properties of the leaves and stem of rice were analyzed at DAS 45? Line 138.
- At what time duration you have analyzed the concentration of N, P, K, and Na in plants and why? Line 149.
- Again, why at 90 DAS you have determined the growth, yield, and grain yield? Reason. Line 158.

Validity of the findings

- 3.77 is pHH20 at what saline concentration? Also where is the average pH value as you have four replicates? Line 180. Also write the standard deviation or error. This implied to whole manuscript.
- Write the exact p value throughout the manuscript. Do not simply say 5%. Make sure add p value everywhere you are talking about significantly different or insignificantly different.
- Add recent references related to effect of nutrients or nitrogen on PNSB to support your findings in discussion section where applicable.
- In conclusion you mentioned that “Those strains should be further applied in the fields where soil is contaminated by salt intrusion in Vietnam. “Line 622-623. Can we not applied these trains in other part of the world facing same salt intrusion?
- In conclusion, please add what can be explored further which you did not explore in your study?

Reviewer 2 ·

Basic reporting

While the study possesses several strengths, there are some significant concerns that need to be addressed through further revision.

Strengths:

Comprehensive Experimental Design: The study demonstrates a well-designed experimental approach, encompassing multiple variables, such as soil characteristics, growth parameters, and yield measurements. This comprehensive approach provides a holistic understanding of the effects of EPS-producing purple non-sulfur bacteria on rice cultivation in acidic salt-affected soil.

Importance of the Research Question: The article addresses a significant issue by investigating the potential benefits of EPS supplementation in mitigating the adverse effects of acidic salt-affected soil on rice growth. This research question holds relevance for agricultural practices in regions facing similar soil challenges.

Concerns:
Redundancy in Data Presentation: The results section contains an overwhelming amount of data presented in multiple tables. While comprehensive data reporting is commendable, it leads to redundancy and hampers the extraction of essential information. Streamlining the tables and focusing on the most critical results would enhance the clarity and cohesiveness of the article.

Text Size and Simplification: The article's text is excessively long and intricate, making it difficult for readers to extract key information. The study would greatly benefit from a substantial reduction in text size and simplification of the language. This revision would improve the article's accessibility and readability, allowing readers to grasp the core findings more effectively.

Experimental design

Lack of EPS Content Discussion: The article consistently references EPS as a crucial factor throughout the text. However, the methods and materials section does not provide any information on EPS content measurement or characterization methods. This omission is a significant drawback, as it prevents a thorough evaluation of the contribution of EPS to the observed effects. Addressing this issue in a revised version is essential.

Include EPS Content Details: To address the major concern, the methods and materials section should provide comprehensive information on EPS content measurement and characterization methods. This addition will enable readers to better understand the role of EPS in the observed effects, thereby strengthening the study's conclusions.

Validity of the findings

The article presents valuable insights into the potential benefits of supplementing EPS-producing purple non-sulfur bacteria on rice growth and soil characteristics in acidic salt-affected soil. However, the lack of EPS content information, excessive text size, and redundancy in data presentation are significant concerns that require revision. Addressing these concerns will greatly improve the article's accessibility, cohesiveness, and impact, providing researchers and practitioners with valuable knowledge for sustainable agricultural practices in challenging soil conditions.

---

## Round 0.2 · Major Revisions

Our apologies for a delayed reply. The authors have Appealed, and we are prepared to consider a revision of the manuscript along the lines of your proposal in your email to us.

Our advice is:

The authors should change the title. For example, it could be:

"Impact of Inoculation with Purple Non-Sulfur Bacteria on Soil Fertility Enhancement and Rice (Oryza sativa) Productivity in Acidic Saline-affected Soils”
or
"Influence of Purple Non-Sulfur Bacterial Augmentation on Soil Nutrient Dynamics and Rice (Oryza sativa) Growth in Acidic Saline-Stressed Environments"

The authors should remove the terms "exopolymeric substance” as there are no evidences to prove the hypothesis.

In addition, one of the Section Editors has expressed the following, which you should address:

"There was a significant time-and-effort placed into the study and the justifications for doing so were straightforward. That being said the suggested amendment to the experimental design may also bring up a valid point. I would be able to see this as acceptable if perhaps the authors can find a way to reflect on their findings and perhaps propose in the discussions the other suggested routes to develop a clearer hint of how the microbial influences may be affecting the results. Perhaps a good discussion of what next steps should be taken to understand the dynamics of this system to improve upon it. There is a high demand for experiments to test plants under stress and this may yield some suggestions to take it to the next level. Perhaps the academic editor might be convinced that this would be a satisfactory approach to not have the efforts go to waste in this case."

· Appeal

Appeal


· · Academic Editor

Reject

While the manuscript has significantly improved, it lacks solid evidence to substantiate the effects of EPS on soil fertility and rice production. The authors have measured indirect parameters and attempted to associate them with EPS.

I recommend that the authors design an experiment to conclusively demonstrate the impact and significance of EPS on soil fertility and rice production. Doing so will align the study more closely with its stated aim and lend substantial weight to the findings.

Therefore, after a careful review of the manuscript and the responses to the reviewers' comments, I am unable to accept your work for publication at this time.

Thank you very much for providing us with the opportunity to consider your work for publication.

·

Basic reporting

1. In the bibliography, references should be numbered to ensure clear differentiation between each source.
2. Blue highlighted color should be removed from the final draft.

Experimental design

No comment.

Validity of the findings

No comment.

Reviewer 2 ·

Basic reporting

All my concerns are cover in the author's review

Experimental design

All my concerns are cover in the author's review

Validity of the findings

All my concerns are cover in the author's review

Additional comments

All my concerns are cover in the author's review

---

## Round 0.3 · accepted · Accept

After careful revision, this work is deemed satisfactory and meets the standards for publication.